# LEARNABLE EIGENFUNCTIONS FOR GRAPH REWIRING: BALANCING LOCAL COMMUNITY STRUCTURE AND GLOBAL CONNECTIVITY

## ABSTRACT

Graph Neural Networks (GNNs) leverage information flow between graph nodes for transductive and inductive tasks. However, default graph topology rarely provides optimal flow for specific tasks, causing over-squashing or over-smoothing. Graph rewiring addresses these issues by altering edges to balance long-range connections (mitigating over-squashing) with locality preservation (preventing over-smoothing). Spectral graph theory offers principled criteria for this trade-off, but has drawbacks: spectral approaches are overly global, and computing spectral quantities lacks scalability. We address these challenges by introducing Inductive Spectral Theory (IST). In IST, spectral quantities and functions are learnable and data-centered, reacting to training data and labels. IST studies spectral elements like the spectral gap and Fiedler vector based on available knowledge. For node and edge-centered tasks, we learn spectral elements from training labels, enabling computation of out-of-sample structural and edgeness measures. This expands structural distances beyond long-range measures like effective resistance to include local intra-cluster-oriented ones. IST is crucial for tasks involving graph populations, such as graph classification, where computing spectral elements is unfeasible, but we learn a consensus spectral space. Our approach strategically adds edges both locally to encourage community structures and globally to facilitate long-range connections while maintaining sparsity. Furthermore, IST serves as a principled graph data augmentation technique, generating diverse training samples that improve model robustness and generalization capabilities. We demonstrate that IST not only improves state-of-the-art graph rewiring performance across multiple benchmarks but also provides a theoretically grounded framework for enhancing GNN architectures through learned spectral properties.

## 1 INTRODUCTION

Graph Neural Networks (GNNs) Yang et al. (2025b); Scarselli et al. (2009); Bruna et al. (2014) have emerged as powerful tools for analyzing graph-structured data, driving significant advancements in social network analysis, molecular biology, and recommendation systems Zhou et al. (2018); Yang et al. (2025a). Most GNN architectures such as Graph Convolutional Network (GCN) Kipf & Welling (2016), Graph Attention (GAT) Veličković et al. (2017) and others Hamilton et al. (2017); Xu et al. (2018) operate through message passing, where node features are iteratively updated by aggregating information from neighboring nodes and generate a new representation (node embeddings) for nodes Gilmer et al. (2017). Further, this node embedding output can perform various tasks like graph and node classification.

However, the GNN's message-passing mechanism faces significant challenges, particularly in practical applications that require capturing long-range interactions. One prominent issue is over-smoothing, where node features become indistinguishable as the number of layers increases Bober et al. (2023); Chen et al. (2024). This convergence of features limits the depth of GNNs, thereby restricting their ability to capture complex relationships within the data. Another critical issue is over-squashing Alon & Yahav (2021), and it occurs when information from an exponentially growing receptive field must be compressed into fixed-size node representations, potentially losing important long-range interactions. Over-squashing is closely related to topological properties of the input graph, such as

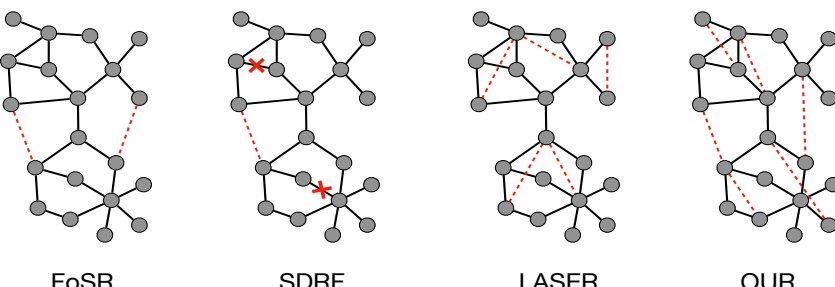

Figure 1: An analysis of various graph rewiring techniques, including FoSR, SDRF, LASER, and IST(our), for mitigating bottlenecks in the input graph.

curvature and effective resistance Arnaiz-Rodríguez et al. (2022); Kedar Karhadkar (2023); Black et al. (2023); Barbero et al. (2023); Attali et al. (2024).

One prevalent strategy to address these issues is graph rewiring, which aims to modify the connectivity of the input graph to improve information flow and alleviate over-squashing. These methods can be broadly categorized into spatial and spectral approaches. Spatial rewiring often focuses on connecting nodes within a certain hop distance, including LASER and hopGNN Gabrielsson et al. (2022); Feng et al. (2022); Barbero et al. (2023) while spectral rewiring optimizes graph-theoretic properties related to connectivity, including Diffwire and First-order Spectral Rewiring (FoSR) Arnaiz-Rodríguez et al. (2022); Kedar Karhadkar (2023); Black et al. (2023) (see Figure. 1). Each approach presents trade-offs between preserving local structure, maintaining sparsity, and enhancing overall graph connectivity.

In addition to over-squashing, GNN models often struggle with limited data for graph classification tasks Zhou et al. (2020), particularly in domains like molecular property prediction where obtaining labeled data is labor-intensive. Data augmentation methods are commonly used to mitigate this issue by adding new features, creating virtual nodes/edges, or generating multiple views of the same graph Rong et al. (2019); Zhou et al. (2020); Zhao et al. (2023); Liu et al. (2025); Wang et al. (2025). This augmentation concept aligns with graph rewiring methods, which improve communication pathways Liu et al. (2022) by strategically adding edges to reduce bottlenecks. This paper proposes inductive spectral theory (IST), a novel graph rewiring method that optimizes graph topology by learning eigenfunctions reactive to graph labels and adding edges locally to encourage community structures and globally to facilitate long-range connections. We utilize the optimized graphs as augmented samples to increase training data size, improving GNN generalization and robustness. We summarize our contributions as follows:

- **Graph Rewiring for Data Augmentation:** We introduce a novel graph rewiring process to generate augmented views of the original graph, effectively increasing both the size and diversity of the training dataset.

- **Label-Reactive Eigenfunction Learning:** Our technique learns eigenfunctions that are reactive to labels, preserving both label information and structural properties of the graph.

- **Multi-scale Edge Addition:** We add edges both locally to encourage community structures and globally to facilitate long-range connections while maintaining graph sparsity to avoid over-smoothing.

- **Over-squashing Mitigation:** Our approach addresses the over-squashing problem common in graph neural networks by introducing strategic long-range connections.

- **Enhanced Model Performance:** These techniques collectively improve model robustness and generalization capabilities through more diverse and representative training samples.

## 2 RELATED WORK

### 2.1 GRAPH REWIRING

Recent research has focused on understanding and mitigating over-squashing in GNN through various approaches. These methods can be broadly categorized into spectral, curvature-based, effective resistance, and locality-aware techniques. *Spectral methods*, such as FoSR by Kedar Karhadkar (2023) aim to improve graph connectivity by maximizing the increase in spectral gap. FoSR adds edges strategically while preserving the original graph structure using a relational GNN architecture. Similar spectral approaches include the work of Banerjee et al. (2022); Yan et al. (2025), who proposed flipping edges based on effective resistance to increase the spectral gap, and Arnaiz-Rodríguez et al. (2022), who developed a method to reweight edges leveraging the Lovász bound. *Curvature-based* approaches leverage the geometric properties of graphs. Nguyen et al. (2023) introduced Batch Ollivier-Ricci Flow (BORF), which uses Ollivier-Ricci curvature to address over-smoothing and over-squashing simultaneously. Their rewiring algorithm modifies local graph geometry to improve information flow. This builds upon earlier work by Topping et al. (2021) who used Forman curvature to analyze over-squashing and proposed a rewiring technique based on increasing edge curvature.

*Effective resistance* methods, exemplified by Black et al. (2023) utilize total effective resistance as a measure of over-squashing. Their approach adds edges to minimize total effective resistance, thereby improving connectivity between all node pairs. This concept is related to the work of Velingker et al. (2023), who proposed incorporating effective resistance-based features into GNNs to capture graph topology information. *Locality-aware* methods, such as Locality-Aware SEquential Rewiring (LASER) by Barbero et al. (2023), attempt to balance local and

Table 1: Properties of different types of rewirings.

| Method | Differentiable | Preserve locality |
|---|---|---|
| FoSR | ✗ | ✗ |
| GTR | ✗ | ✗ |
| LASER | ✗ | ✔ |
| Diffwire | ✔ | ✗ |
| Ours (IST) | ✔ | ✔ |

global graph properties. LASER uses a sequence of rewiring operations considering connectivity measures and locality constraints, aiming to preserve graph sparsity and local structure while reducing over-squashing. This approach shares similarities with multi-hop aggregation methods proposed by Abu-El-Haija et al. (2019) and Wang et al. (2021), which also attempt to capture local and global graph information. These diverse approaches to graph rewiring offer various strategies for mitigating over-squashing: FoSR adds edges based on spectral properties, BORF modifies edge weights to increase curvature, effective resistance methods add edges to minimize total resistance, and LASER uses a sequential process balancing local and global connectivity improvements. Each method provides unique insights into addressing the over-squashing problem while attempting to preserve important graph properties (see Table 1).

### 2.2 GRAPH AUGMENTATION

Data augmentation methods aim to improve the generalization and robustness of deep neural networks, particularly in fields such as computer vision (CV) Shorten & Khoshgoftaar (2019) and natural language processing (NLP) Zhang et al. (2015). In CV, methods such as image flipping, noise injection, and cutout have been widely adopted to generate more varied training datasets. Similarly, generative models like Variational Auto-Encoders (VAEs) Kingma & Welling (2013) and Generative Adversarial Networks (GANs) Goodfellow et al. (2014) can produce new samples by learning the underlying data distribution. However, applying data augmentation techniques to graph-structured data presents unique challenges due to the non-Euclidean nature of graphs, where nodes are irregularly connected by edges Zhao et al. (2022); Ding et al. (2022); Guo et al. (2025); Liu et al. (2024).

Recent works have focused on developing graph augmentation by revising the graph's structures and manipulating node features for node-level and graph-level prediction tasks Verma et al. (2021); Kong et al. (2020). Feature-based augmentation methods manipulate node features to create new training samples. Researchers have recently developed Mixup augmentation methods Verma et al. (2021); Han et al. (2022) for graph augmentation, which generates augmented graphs by interpolating the features of node pairs or through adversarial learning. However, the most commonly used graph enhancement methods are based on the random modification of graph structures, where edges and nodes are added or removed randomly Rong et al. (2019); You et al. (2020); Zhou et al. (2020). Such random transformations may destroy the original topological structural characteristics of the graph

and alter label-related information, potentially reducing the effectiveness of these augmentations for improving graph classification model performance Rong et al. (2019).

Our approach of graph rewiring strategically improves the communication pathways within a graph by adding edges to reduce bottlenecks: it is both local and global. This results in a new, optimized graph structure that addresses the over-squashing issue and serves as an augmented view of the graph. By using this rewired graph as an augmented sample, we can increase the size and diversity of the training data, thereby enhancing the model's robustness and generalization capabilities.

## 3 PRELIMINARIES

### 3.1 GRAPH NEURAL NETWORKS

Graph Neural Networks (GNNs) are specialized deep learning models for data represented as graphs. GNNs operate on the principle of message passing, where nodes iteratively update their states by integrating information from their neighbors. Formally, for layer $l$ in a GNN, the representation of node $v$ in the next layer $h_v^{(l+1)}$ is computed as follows:

$$h_v^{(l+1)} = \sigma \left( \sum_{u \in N(v)} \mathbf{A}_{vu} \cdot \mathbf{W}^{(l)} h_u^{(l)} \right), \tag{1}$$

where $N(v)$ denotes the set of neighbors of node $v$, $\mathbf{A}$ is the adjacency matrix, $\mathbf{W}^{(l)}$ is a learnable weight matrix for layer $l$, $h^{(l)}$ is the matrix of node features at layer $l$, and $\sigma$ is a nonlinear activation function.

The Graph Isomorphism Network (GIN) Xu et al. (2019) has emerged as a powerful variant within GNNs, known for its ability to differentiate between non-isomorphic graph structures. GIN employs an aggregation function defined as:

$$h_v^{(l+1)} = \text{MLP} \left( (1 + \epsilon) \cdot h_v^{(l)} + \sum_{u \in N(v)} h_u^{(l)} \right), \tag{2}$$

where $\epsilon$ is a learnable parameter and MLP stands for a multi-layer perceptron. This approach ensures that GIN robustly captures graph structures by flexibly combining the central node's information with that of its neighbors.

In recent years, GIN-based networks have demonstrated high efficacy in various tasks, including graph classification, link prediction, and community detection Hoseinnia et al. (2025). They effectively utilize graph topological information and local node features, offering a potent and adaptable method for handling graph-structured data with strong predictive capabilities and generalization.

### 3.2 SPECTRAL GRAPH THEORY

In a graph $G = (V, E)$ with $N = |V|$ nodes and edges $|E|$, with $E \subseteq V \times V$ the adjacency matrix $\mathbf{A} \in \{0, 1\}^{N \times N}$ is a square matrix where $\mathbf{A}_{ij} = 1$ if edge $(i, j) \in E$, and $0$ otherwise. The degree matrix $\mathbf{D}$ is a diagonal matrix with $d_i = \mathbf{D}_{ii}$ representing the degree of node $i$, which is the count of edges connected to $i$. Then, from $\mathbf{A}$ and $\mathbf{D}$ we obtain the *graph Laplacian* $\mathbf{L} := \mathbf{D} - \mathbf{A}$. The *normalized Laplacian* $\mathcal{L}$ is given by $\mathcal{L} := \mathbf{I} - \mathbf{D}^{-\frac{1}{2}} \mathbf{A} \mathbf{D}^{-\frac{1}{2}}$, where $\mathbf{I}$ is the identity matrix. The eigenvalues of the Laplacian and the normalized Laplacian offer insights into various structural aspects of the graph, including connectivity, community structure, and information diffusion. Specifically, the spectrum of the normalized Laplacian $\mathcal{L}$ consists of non-negative real numbers ordered as $0 = \lambda_1 \leq \lambda_2 \leq \cdots \leq \lambda_n \leq 2$. Given the spectrum and the corresponding eigenvectors $\mathbf{u}_i \in \mathbb{R}^N$ satisfying $\mathcal{L} \mathbf{u}_i = \lambda_i \mathbf{u}_i$, the spectral decomposition of $\mathcal{L}$ is given by $\mathcal{L} = \mathbf{U} \text{diag}(\lambda_1, \lambda_2, \ldots, \lambda_n) \mathbf{U}^T = \sum_i \lambda_i \mathbf{u}_i \mathbf{u}_i^T$.

Spectral Graph Theory (SGT) Chung (1997) addresses the study of the normalized Laplacian's spectra and their eigenvectors. The most important of these vectors is $\mathbf{v}_2$ (the *Fiedler vector*) the one associated with the *spectral gap* $\lambda_2$ (which is positive if the graph is connected). The gap is a fundamental quantity in SGT (e.g. it bounds the graph connectivity and its inverse determines the

mixing time of random walks). It is obtained as follows:

$$\lambda_2 = \min_{f \perp \mathbf{D}^{1/2}\mathbf{1}} \frac{\mathcal{E}(f)}{\sum_{i \in V} f_i^2 d_i} = \min_f \frac{\text{vol}G \cdot \mathcal{E}(f)}{\sum_{i,j} (f_i - f_j)^2 d_i d_j} \; . \tag{3}$$

where $\perp$ stands for perpendicular, $\text{vol}G = \sum_{i \in V} d_i$ is the volume of the graph and $\mathcal{E}(f) := \sum_{i \sim j} (f_i - f_j)^2$ is known as the *Dirichlet energy* of $f : V \to \mathbb{R}^N$. Actually $\mathbf{u}_2 = \mathbf{D}^{1/2} f$. Herein it is key to note that $f \perp \mathbf{D}^{1/2}\mathbf{1}$ where $\mathbf{u}_1 = \mathbf{D}^{1/2}\mathbf{1}$.

One key concern in this paper is *spectral clustering*. It is well known Shi & Malik (2000)von Luxburg (2007) that

$$\frac{\mathcal{E}(f)}{\sum_{i \in V} f_i^2 d_i} \leq \text{Ncut}(A, B) := \frac{\text{cut}(A, B)}{\text{vol}A} + \frac{\text{cut}(A, B)}{\text{vol}A} \; , \tag{4}$$

where $V = A \cup B$, $A \cap B = \emptyset$ is the optimal partition in terms of minimizing the normalized cut Ncut(A,B), which is an NP-Hard problem. In general, if we pack the $K$ smallest eigenvectors of $\mathcal{L}$ in a $N \times K$ matrix $\mathbf{U}$, feeding a K-means clustering with the rows of this matrix leads to partitioning the graph into $K$ communities $C_1, C_2, \ldots, C_K$. Interestingly, the squared distances between two rows are bounded as follows Hofmeyr (2020):

$$\left\| \frac{\mathbf{U}_{i,1:K}}{\sqrt{d_i}} - \frac{\mathbf{U}_{j,1:K}}{\sqrt{d_j}} \right\|^2 \leq \max_M NK \cdot \text{NCut}(C_M, C_{L \neq M}) \; . \tag{5}$$

Therefore, small distances between the rows in $\mathbf{U}$ are usually associated with nodes in the same cluster and larger distances correspond to inter-cluster nodes.

## 4 METHODOLOGY

Despite the usefulness of SGT for providing a wide catalog of topologically meaningful distances (both local and global) to rewire a graph, the computational cost of computing the eigenvectors is $O(N^3)$. This is not feasible for large graphs. In addition, in some tasks such as graph classification (see below), where several training graphs per class are provided, SPG is limited. It cannot capture the typical eigenvectors of each class or find a consensus eigenspace for all classes.

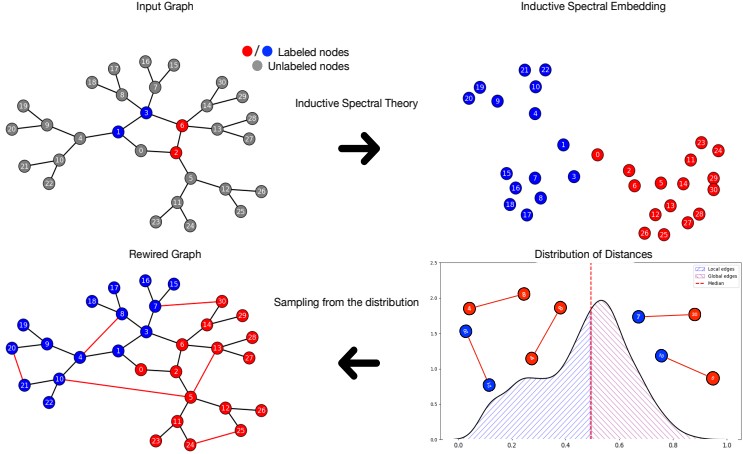

Figure 2: Visualization of the IST process for graph rewiring. The figure illustrates the transformation from an input graph to a rewired graph through IST. It shows how labeled and unlabeled nodes are mapped to an inductive spectral embedding, resulting in a distribution of distances. The rewired graph is then created by sampling from this distribution, adding both local and global edges based on the learned spectral properties. When applied to node classification, this input graph induces over-squashing but this is avoided by clustering the node embeddings. Over-smoothing is reduced by increasing $\mathcal{E}$ guided by $\mathcal{L}_{task}$.

## 4.1 Inductive Spectral Theory

IST studies the expressiveness of the spectral elements of $\mathcal{L}$ (eigenfunctions, gaps and distances) derived from

$$\min_{f \perp \mathcal{P}} \frac{\mathcal{E}(f)}{\sum_{i \in V} f_i(\mathbf{A})^2 d_i} + \mathcal{L}_{task} \ . \tag{6}$$

Firstly, $\mathcal{E}(f) := \sum_{i \sim j}[f_i(\mathbf{A}) - f_j(\mathbf{A})]^2$ is a Dirichet energy where $f_i$ and $f_j$ are scalars, the components of a learnable mapping $f : \mathbf{A} \to \mathbb{R}^N$. The purpose of $f$ is to leverage high-order (HO) similarities (common neighbors, see below) between $\mathbf{a}_{:i}$ and $\mathbf{a}_{j:}$, the columns in $\mathbf{A}$ of the nodes in $V$ linked by the edges $(i, j) \in E$. Then, the eigenvectors $f$ which are the natural minimizers of the Dirichlet energies incorporate these similarities in a catalog of orthogonal functions with respect to an eigenspace $\mathcal{P}$.

On the other hand, $\mathcal{L}_{task}$ is the task-dependent classification loss. Node classification, graph classification, and link prediction are downstream tasks where IST may leverage partially-observed labels to find data-centered eigenvalues and eigenvectors.

IST is rooted in structural semi-supervised learning Song et al. (2023), but herein we incorporate the recent trend in large graph mining where $f(\mathbf{A})$ is an MLP Lim et al. (2021). Making this MLP reactive to the task loss $\mathcal{L}_{task}$, i.e. minimizing $\mathcal{E}(f(\mathbf{A})) + \mathcal{L}_{task}$, we transfer the training labels to the learning of eigenvectors.

**Common Neighbors**. IST exploits the following observation. Given $f(\mathbf{A}) \in \mathbb{R}^{K \times N}$, with $f(\mathbf{A}) = \sigma(\mathbf{WA})$, and the learnable weight matrix $\mathbf{W} \in \mathbb{R}^{K \times N}$, the expansion $(\mathbf{WA})_{ip} = \sum_{p \in N(i)} \mathbf{W}_{ip}$ means that if a node $i$ has many neighbors $p$ of a given community, then they $i$ and $p$ belong to the same community and $\mathbf{W}_{ip}$ will be large on average. This is consistent with the *friendship paradox* (my friends have more friends than me). Therefore, for the general model $f(\mathbf{A}) = \text{MLP}_\theta(\mathbf{A})$, the extension of Eq. 6 for computing all the *empirical eigenfunctions* (EE) is

$$\min_\theta \text{Trace}[f(\mathbf{A})^T \mathcal{L} f(\mathbf{A})] + \mathcal{L}_{task} \ \text{s.t.} \ f(\mathbf{A})f(\mathbf{A})^T = \mathbf{I} \ . \tag{7}$$

We solve such a problem via SGD. Denoting a generic column of $f(\mathbf{A})$, such as the Fiedler vector $f$, we have characterized its structure in terms of the weights of the MLP. For a single-layer MLP we prove that such a structure *is dominated by the number of common neighbors* (**Theorem 1** and its corollaries in the Supplementary). To give here an intuition about this fact, note that $f_i = \sigma(\mathbf{W}_{i,:}\mathbf{A})$ with $\sigma = \tanh$ for providing bipolar outputs. Then the Dirichlet energy $\sum_{i \sim j}(g_i - g_j)^2$ of the respective logits $g_i := \mathbf{W}_{i,:}\mathbf{A}$, $g_j := \mathbf{W}_{j,:}\mathbf{A}$ is expanded as follows:

$$\sum_{i \sim j}(g_i - g_j)^2 = \sum_{i \sim j}[\sum_{p \in N(i)} \mathbf{W}_{ip} - \sum_{q \in N(j)} \mathbf{W}_{jq}]^2 \ , \tag{8}$$

where the weights corresponding to the common neighbors $r \in N(p) \cap N(q)$ are included (if they do exist). Note that now we are comparing neighborhoods and their weights instead of scalars as in $\sum_{i \sim j}(f_i - f_j)^2$ which is combinatorially richer. The role of common neighbors allows us to study the particularities of trees vs graphs (with cycles).

**Transductive/Inductive Power**. Given that we learn a non-linear mapping $MLP_\theta(\mathbf{A})$, we can perform both transductive and inductive learning. For instance, when the task is node classification we can either predict the labels of test nodes or analyze the robustness of the model under structural attacks. Link prediction is more inductive and common-neighbors heuristics usually drives it. Finally, graph classification has been usually addressed via transductive methods, but in this paper, we show how to provide out-of-the-sample graphs via structural data augmentation.

Overall, the number of labeled samples needed to achieve a good generalization performance *depends on the degree distribution*. We cover this issue in **Theorem 2** and its corollaries in the Supplementary. Again, to give an intuition, note that the denominator of Eq. 6 as per the logits $g_i$ can be expanded as follows:

$$\sum_{i \in V} g_i^2 d_i = \sum_{i \in V}[\sum_{p \in N(i)} \mathbf{W}_{ip}]^2 d_i = \sum_{i \in V} \mathbf{W}_{ip}^2 \sum_{p \in N(i)} d_p \ . \tag{9}$$

Since the denominator is maximized, the magnitude of the weights increases proportionally to $\sum_{p \in N(i)} d_p$ instead of $d_i$ as in Eq. 6. This results in more separable weights thus avoiding close-to-zero entries in the Fiedler vector whenever $d_i$ is large enough. In general, large degrees lead to a

small number of labeled samples. In the Supplementary, we will also provide extensive experiments with different types of graphs (trees, SBMs, cycles, etc).

## 4.2 METHOD: GRAPH CLASSIFICATION

Following IST, graph classification is addressed as follows.

**1) Consensus EEs**. Given a set of training samples $\mathcal{T} = \{(G_i, l_i)\}$ (graphs and labels), we feed an MLP with the adjacencies $\{\mathbf{A}_i\}$ (padding ensures a common size) and labels $\{l_i\}$: $f_1(\mathbf{A}) = \text{MLP}_1(\{\mathbf{A}_i, l_i\})$ minimizes the loss Trace$+\mathcal{L}_{task}$ in Eq. 7 and $f_1(\mathbf{A})$ encodes a *consensus eigenspace* of $K$ EEs. $K$ is a hyperparameter.

**2) Mapping**. We train a second MLP, with $f_1(\mathbf{A})$ and the labels. Actually, we have

$$\mathbf{Z} = \text{Readout}(\text{MLP}_2(\text{MLP}_1(\{\mathbf{A}_i, l_i\}))) , \qquad (10)$$

where the second MLP maps $f_1(\mathbf{A})$ with $K$ eigenvectors to $f_2(\mathbf{A})$ with $\mathcal{C}$ (number of classes) eigenvectors. Finally, Readout is a permutation-invariant operation that combines the representations of the nodes (rows of $f_2(\mathbf{A})$).

**3) Nodal distances**. Now, we freeze the weights of $\text{MLP}_1$ and we feed it with the training adjacencies $\{\mathbf{A}_i\}$. Each of the predicted eigenspaces $\hat{f}(\mathbf{A}_i)$ provides a distribution of pairwise distances $\mathcal{D}_i$ between the rows of the predicted eigenspace associated with the nodes of $G_i = (V_i, E_i)$.

**4) Data augmentation**. We augment the edges of each training graph $G_i = (V_i, E_i)$ by sampling $\mathcal{D}_i$ for adding $N/2$ *local* edges, and $N/2$ *global* ones, where $N$ is the common padding size. We add a local edge if the distance between its nodes $(i, j)$ is smaller or equal to the median (see Figure 2). Otherwise, we have a global edge.

**5) GNNs**. We train the GNNs both with the original $\{G_i\}$ and augmented $\{\tilde{G}_i\}$ graphs. Then we perform the test and provide the accuracy. In graph classification, the label transfer is not as obvious as in node classification. Note that the colored labels in the input graph of Figure 2 are induced by the weights of the MLP when they react to $\mathcal{L}_{task}$.

## 4.3 COMPUTATIONAL EFFICIENCY

The computational complexity of IST is primarily determined by the learning of eigenfunctions and the subsequent rewiring process. For a graph with $N$ nodes and $E$ edges, the space complexity of our method is $O(NK)$, where $K$ is the number of learned eigenfunctions. The time complexity for computing the Dirichlet energy and task-specific loss is $O(EK + NK^2)$, leveraging sparse matrix operations for efficiency. The edge addition step, both local and global, has a complexity of $O(N \log N)$ due to the use of efficient sampling techniques. Overall, IST's computational cost scales favorably with graph size, making it applicable to large-scale graph learning tasks. This efficiency is particularly noteworthy when compared to traditional spectral methods that often require $O(N^3)$ operations for eigendecomposition. Compared to other state-of-the-art methods, IST demonstrates competitive computational efficiency: GTR requires $O(N^3)$ operations, BORF scales as $O(Ed^3)$ (where $d$ is the maximal degree), LASER needs $O(N^3)$, and SDRF has $O(N^2)$ complexity. With IST's complexity of $O(EK + NK^2)$, where typically $K \ll N$, our approach offers a more scalable alternative for graph rewiring and data augmentation in the context of GNNs.

## 5 EXPERIMENTS

This section provides an empirical evaluation of IST's effectiveness across various tasks, such as node classification and graph classification, in comparison to other rewiring techniques like curvature-based methods, spectral gap approaches, and locality-aware strategies. The code used for these experiments can be found at [1].

**Datasets:** We conduct experiments on a range of standard node and graph classification tasks, following the same methodology as BORF Nguyen et al. (2023) to ensure a fair comparison.

---

[1] https://anonymous.4open.science/r/IST-6056

Table 2: Comparison of the proposed method and baselines. The bold numbers represent the highest accuracy score, and OOR is referred to as out-of-resource.

| Classification | Methods | Mutag | BZR | Mutagen | PTCMM | PROTEINS | ENZYMES | IMDB-B | COLLAB |
|---|---|---|---|---|---|---|---|---|---|
| None | GIN | 76.02±0.03 | 79.45±0.01 | 79.59±0.03 | 62.05±0.01 | 69.23±0.01 | 30.25±0.01 | 67.12±0.01 | 71.77±0.04 |
| Rewiring | SDRF | 78.10±0.02 | 80.20±0.01 | 79.75±0.03 | 59.08±0.01 | 70.31±0.01 | 31.30±0.02 | 67.10±0.01 | 73.20±0.04 |
| | FOSR | 74.62±0.02 | 79.50±0.01 | 79.10±0.03 | 60.45±0.01 | 72.41±0.08 | 24.10±0.01 | 66.30±0.09 | 73.01±0.04 |
| | GTR | 79.45±0.02 | 80.58±0.02 | 79.89±0.02 | 61.45±0.02 | 70.17±0.01 | 29.01±0.01 | 67.21±0.02 | OOR |
| | DiffWire | 75.21±0.02 | 78.34±0.01 | 79.09±0.02 | 62.17±0.02 | 69.25±0.01 | 28.03±0.01 | 68.30±0.03 | 73.78±0.04 |
| | BORF | 77.30±0.02 | 79.45±0.01 | OOR | 63.25±0.01 | 69.75±0.08 | 29.76±0.01 | 67.35±0.09 | OOR |
| | LASER | 72.95±0.02 | 78.58±0.01 | 61.48±0.01 | 59.25±0.02 | 63.77±0.19 | 20.73±0.08 | 69.07±0.09 | 72.50±0.04 |
| Augmentation | DropEdge | 77.58±0.61 | 79.75±0.57 | 78.08±0.19 | 62.82±0.61 | **74.31±0.27** | 31.83±0.61 | 64.90±0.47 | 60.90±4.47 |
| | DropNode | 78.80±0.85 | 79.87±0.48 | 77.50±0.31 | 56.21±0.61 | 72.77±0.53 | 31.54±0.54 | 68.50±0.59 | 68.50±0.47 |
| | M-Evolve | 75.59±0.94 | 79.30±0.51 | 77.84±0.18 | 58.75±0.71 | 72.31±0.38 | 32.35±0.61 | 67.40±0.67 | 61.50±0.71 |
| | Gmixup | 78.10±0.65 | 80.89±0.42 | 78.08±0.64 | 62.30±0.68 | 65.81±2.13 | 30.66±4.39 | 68.10±1.25 | 73.10±0.59 |
| **Ours** | **IST** | **81.20±0.02** | **81.02±0.01** | **80.69±0.03** | **66.01± 0.01** | 70.57±0.08 | **34.68±0.01** | **69.10±0.01** | **75.39±0.04** |

Table 3: Comparison of the proposed method and baselines. The bold numbers represent the highest accuracy score.

| | GCN | | | | | GIN | | | | |
|---|---|---|---|---|---|---|---|---|---|---|
| | None | SDRF | FoSR | BORF | IST | None | SDRF | FoSR | BORF | IST |
| **Cora** | 86.7 ± 0.3 | 86.3 ± 0.3 | 85.9 ± 0.3 | 87.5 ± 0.2 | **88.1 ± 0.3** | 76.0 ± 0.6 | 74.9 ± 0.1 | 75.1 ± 0.8 | 78.4 ± 0.4 | **78.6 ± 0.3** |
| **Citeseer** | 72.3 ± 0.3 | 72.6 ± 0.3 | 72.3 ± 0.3 | 73.8 ± 0.2 | **74.1 ± 0.2** | 59.3 ± 0.9 | 60.3 ± 0.8 | 61.7 ± 0.7 | 63.1 ± 0.8 | **63.4 ± 0.4** |
| **Texas** | 44.2 ± 1.5 | 43.9 ± 1.6 | 46.0 ± 1.6 | 49.4 ± 1.2 | **52.4 ± 1.0** | 53.5 ± 3.1 | 50.3 ± 3.7 | 47.0 ± 3.7 | 63.1 ± 1.7 | **66.9 ± 1.3** |
| **Cornell** | 41.5 ± 1.8 | 42.2 ± 1.6 | 40.2 ± 1.6 | **50.8 ± 1.1** | 50.1 ± 0.9 | 36.5 ± 2.2 | 40.0 ± 2.1 | 35.6 ± 2.4 | **48.6 ± 1.2** | 48.4 ± 1.8 |
| **Wisconsin** | 44.6 ± 1.4 | 46.2 ± 1.2 | 48.3 ± 1.3 | 50.3 ± 0.9 | **51.1 ± 0.7** | 48.5 ± 2.2 | 48.8 ± 1.9 | 48.5 ± 2.1 | 54.9 ± 1.2 | **56.0 ± 1.1** |
| **Chameleon** | 59.2 ± 0.6 | 59.4 ± 0.5 | 59.3 ± 0.6 | 61.5 ± 0.4 | **62.0 ± 0.5** | 58.1 ± 2.1 | 58.4 ± 2.1 | 56.3 ± 2.2 | 65.3 ± 0.8 | **66.8 ± 1.3** |

For node classification, we report our findings using datasets such as Cora, Citeseer Sen et al. (2008), Texas, Cornell, Wisconsin Pei et al. (2020), and Chameleon Rozemberczki et al. (2019), comparing BORF against both the baseline of no graph rewiring and two other rewiring techniques.

For graph classification, we evaluate well-established benchmarks like PROTEINS, ENZYMES, COLLAB, MUTAG, and IMDB-BINARY Morris et al. (2020), which are known for requiring long-range interactions as discussed in Kedar Karhadkar (2023). Additionally, we incorporate three more datasets—BZR, PTCMM, and MUTAGENICITY Zhou et al. (2020)—to further assess the effectiveness of our approach, particularly in scenarios involving varied dataset sizes and complexities. More detailed information about all the datasets used can be found in the Supplementary.

**Baselines:** For graph classification, we benchmark IST against several state-of-the-art rewiring approaches. These include no graph rewiring as a baseline, SDRF Topping et al. (2021), which leverages discrete Ricci curvature for graph rewiring, and BORF Nguyen et al. (2023), another curvature-based rewiring method. We also compare against FoSR Kedar Karhadkar (2023), which optimizes the spectral gap of the graph, and Locality-aware LASER Barbero et al. (2023), which focuses on preserving local structure during rewiring. These comparisons aim to verify the efficiency and effectiveness of our IST method across various graph structures. To further assess the performance of IST in an augmentation setting for graph classification, we extend our evaluation to include several widely used graph augmentation techniques. These include DropEdge Rong et al. (2019), which randomly removes a certain fraction of edges from the input graph, and DropNode You et al. (2020), which randomly removes nodes and their associated edges. We also consider M-Evolve Zhou et al. (2020), which generates new graphs through a graph evolution process, and Gmixup Han et al. (2022), which creates new graphs by interpolating between existing ones. For node classification tasks, we specifically focus on rewiring methods that have proven effective in this context. We compare IST with a baseline without rewiring, SDRF Topping et al. (2021), FoSR Kedar Karhadkar (2023), and BORF Nguyen et al. (2023). These methods represent some of the few approaches that have addressed node classification through graph rewiring, making them crucial baselines for our evaluation. While many other methods exist in the state-of-the-art for node classification, we specifically concentrate on those employing rewiring techniques to maintain consistency with our approach.

**Experiment setup:** For graph classification, augmentation techniques are implemented as preprocessing steps on training datasets, with results evaluated using GCN and GIN architectures. We use consistent hyperparameters: 64 hidden units, 0.5 dropout rate, 4 layers, learning rate 0.001, weight decay 0.00001, maximum 1000 epochs with early stopping after 100 epochs without improvement, conducting 100 random trials for robustness. For node classification, we follow BORF's experimental setup with 10 runs per experiment, 60/20/20 train/validation/test split, and use BORF's suggested conditions across all methods without hyperparameter tuning to ensure fair comparison.

**Results:** Our comprehensive evaluation demonstrates IST's exceptional efficacy across both graph and node classification tasks. In graph classification (Table 2), IST consistently achieves superior accuracy compared to existing rewiring and augmentation techniques across molecular, bioinformatics, and social network datasets, yielding an average improvement of 2.0% with the sole exception being the proteins dataset. For node classification (Table 3), IST demonstrates remarkable consistency, achieving the highest accuracy scores across all evaluated datasets for both GCN and GIN architectures, notably reaching 88.1% and 78.6% respectively on the Cora dataset. Unlike methods such as SDRF that rely solely on local curvature or FoSR/GTR that may add edges indiscriminately, IST employs a balanced strategy considering both local and global graph properties, introducing edges that enhance structural cohesion and optimize information flow.

**Ablation Study:** To dissect the contributions of various components within IST, we conducted an ablation study across four representative graph classification datasets of varying sizes. We examined the impact of local edge addition (IST w/o Local), global edge addition (IST w/o Global), augmentation (IST w/o augmentation), and label information in eigenfunctions (IST w/o Label). The results, presented in Table 4, offer valuable insights into the method's efficacy. Our findings reveal that the optimal edge addition strategy varies depending on the dataset characteristics. For instance, local edge addition within communities proved most beneficial for Enzymes and PTCMM datasets, while global edge addition for enhanced long-range connections was superior for Mutag and IMDBB. This variability underscores the importance of IST's adaptive approach in addressing dataset-specific structural needs. Furthermore, the augmentation component of IST demonstrated significant performance enhancements, particularly on smaller datasets such as Mutag, Enzymes, and PTCMM. This observation highlights the crucial role of IST in mitigating over-squashing effects, thereby improving the overall performance of GNN models across diverse graph structures.

Table 4: Ablation studies about different IST components.

| Architecture | Mutag | ENZYMES | PTCMM | IMDB-B |
|---|---|---|---|---|
| IST | **81.20 ±0.02** | **34.68±0.01** | **66.01 ± 0.01** | **69.10± 0.01** |
| IST w/o Local | 80.45±0.02 | 33.76±0.01 | 64.64±0.02 | 68.52±0.01 |
| IST w/o Global | 80.07±0.02 | 34.21±0.01 | 65.64±0.01 | 68.38±0.01 |
| IST w/o Augmen | 77.39±0.02 | 33.03 ±0.01 | 63.22±0.01 | 69.06±0.01 |
| IST w/o Label | 80.85±0.02 | 34.11±0.01 | 64.94± 0.02 | 69.03±0.02 |

# 6 CONCLUSION

In this paper, we have introduced Inductive Spectral Theory (IST) as a novel approach to address the limitations of traditional graph rewiring techniques in GNNs. By making spectral quantities and functions learnable (e.g., eigenfunctions), IST provides a data-centered framework that adapts to the specific requirements of node, edge, and graph-level tasks. Our approach mitigates common issues such as over-squashing and over-smoothing by balancing long-range connectivity and locality and enhances scalability and applicability in diverse contexts, including graph classification. Furthermore, IST offers a principled methodology for graph data augmentation, pushing the boundaries of current graph rewiring techniques. Our results demonstrate that IST advances state-of-the-art graph rewiring and establishes a robust foundation for future research in graph-based learning tasks.

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

## A  Summary of Results

The results below explore the expressive power of the learnable weights $\mathbf{w}$. Each component of the Fiedler vector $f$ is encoded as $f_i = <\mathbf{w}, \mathbf{a}_{:i}> = \sum_{p \in N(i)} \mathbf{w}_p$, i.e. a projection of the corresponding column in the adjacency matrix (permutation-invariant).

**Theorem 1** shows that the Fiedler vector $f$ and, consequently, the spectral gap $\lambda_2$ can be expressed in terms of common neighbors.

**Corollary 1** shows that the denser the graph the closer the learnable weights $\mathbf{w}$ to the Fiedler vector provided by the standard Spectral Graph Theory (SGT). This is enabled by the large amount of common neighbors arising in dense graphs.

**Corollary 2** reveals that graph cuts are relaxed in IST with respect to their counterparts in SGT. This is due to the second-order constraints (neighbor of neighbor) imposed on the weights.

**Corollary 3**. Notable nodes tend to have larger components in terms of their magnitude $\mathbf{w}_i^2$.

**Corollary 4**. Extremal nodes in paths or leaves in trees (unit degree) tend to have small $\mathbf{w}_i^2$.

**Corollary 5**. However, when these extremal nodes and leaves are linked to preceding nodes in the structure (path or tree) their weight magnitude becomes similar to that of the nodes in the same loop. In other words, loops smooth magnitudes.

**Lemma 1** shows that the weight vector must be orthogonal to the vector of local volumes: $\mathbf{w} \perp \mathbf{d}_N$. This explains the usual condition $f \perp \mathbf{D1}$. However, the new condition has deeper implications since local volumes are typically larger than individual degrees. This also explains cut relaxation with respect SGT.

**Lemma 2** adapts the Harnack equality to explain label diffusion. In other words, the Fiedler vector is a harmonic function (the value of a component is the neighboring average). This results in weights and labels being related by the inverse of the degree.

**Theorem 2** Leverages Lemma 2 to show that label diffusion leads to uncertainty (components $f_i \approx 0$) as the unlabeled node is far from the labeled one in terms of shortest paths.

**Corollary 6** is a "positive" version of Theorem 2: large degrees favor label propagation. Therefore, there is a trade-off between small degrees which limit uncertainty, and large degrees which favor label propagation. In other words, the degree distribution drives label propagation.

**Corollary 7** leverages Lemma 2 to show the need for both positive and negative labels. In other words, in the absence of negative labels, the available information on positive labels overrides the min-cut principles of standard SGT in most of cases.

## B    RESULTS

**Theorem 1.** *For a linear mapping $f_i = <\mathbf{w}, \mathbf{a}_{:i}>$, where $\mathbf{a}_{:i}$ is the $i-$th column of $\mathbf{A}$ and $\mathbf{w} \in \mathbb{R}^N$ is a learnable vector, the IST spectral gap $\lambda_2$ is dominated by the maximal number of common neighbors.*

*Proof.* Following Chung[2], the spectral gap $\lambda_2$ is given by

$$\lambda_2 = \min_{f \perp \mathbf{D1}} \frac{\sum_{i \sim j}(f_i - f_j)^2}{\sum_{i \in V} f_i^2 d_i} \ , \tag{11}$$

where $d_i$ denotes the degree of node $i$. From the fact that

$$f_i = <\mathbf{w}, \mathbf{a}_{:i}> = \sum_{p \sim i} \mathbf{w}_p \tag{12}$$

we obtain

$$\sum_{i \sim j}(f_i - f_j)^2 = \sum_{i \sim j}[\sum_{p \in N(i)} \mathbf{w}_p - \sum_{q \in N(j)} \mathbf{w}_q]^2 \ . \tag{13}$$

Then, we proceed to rewrite the denominator as

$$\sum_{i \in V} f_i^2 d_i = \sum_{i \in V} \mathbf{w}_i^2 \sum_{p \in N(i)} d_p \tag{14}$$

Then,

$$\lambda_2 = \min_{f \perp \mathbf{D1}} = \frac{\sum_{i \sim j}[\sum_{p \in N(i)} \mathbf{w}_p - \sum_{q \in N(j)} \mathbf{w}_q]^2}{\sum_{i \in V} \mathbf{w}_i^2 \sum_{p \in N(i)} d_p} \tag{15}$$

which uncovers the second-order constraints on the components $\mathbf{w}_i$ leading to the Fiedler vector $f$. Then, expanding the numerator, the structure of each term $i \sim j$ is given by

$$\sum_{i \sim j}(f_i - f_j)^2 = [\sum_{U(p)} \mathbf{w}_p - \sum_{U(q)} \mathbf{w}_q + (\mathbf{w}_j - \mathbf{w}_i)]^2 \tag{16}$$

Where $U(p) = \{p \in N(i), p \neq j, p \notin CN_{ij}\}$ and $U(q) = \{q \in N(j), q \neq i, q \notin CN_{ij}\}$, where $CN_{ij}$ is the set of *common neighbors* of nodes $i$ and $j$. As a result, the existence of common neighbors determines the structure of the IST Fiedler vector. $\qquad\square$

---

[2]Fan R.K. Chung: Spectral Graph Theory, AMS, 1994.

**Corollary 1.** *For the complete graph (clique) $K_N$, with $N > 2$, the IST Fiedler vector is coincident with that of the standard Spectral Theory and it is mirrored by the optimal weights $\mathbf{w}$.*

*Proof.* For $K_N$ every node has $N - 1$ neighbors and $N - 2$ common neighbors for each edge $i \sim j$. As a result, in Eq. 16 we have that $U(p) = U(q) = \emptyset$ for all edges.

$$\lambda_2 = \min_{f \perp \mathbf{D1}} = \frac{\sum_{i \sim j}(\mathbf{w}_i - \mathbf{w}_j)^2}{\sum_{i \in V} \mathbf{w}_i^2 \sum_{p \in N(i)} d_p} \ . \tag{17}$$

Since in $K_N$ any node $i$ is linked with any other $p \neq i$, we have

$$\sum_{p \in N(i)} d_p = \sum_{p \in V} d_p = \text{vol}G \ , \tag{18}$$

which is a constant.

Therefore, the IST Fiedler's vector and value for $K_N$ are almost equal to those provided by the standard Spectral Theory:

$$\lambda_2 = \min_{\mathbf{w} \perp \mathbf{D1}} \frac{\sum_{i \sim j}(\mathbf{w}_i - \mathbf{w}_j)^2}{\sum_{i \in V} \mathbf{w}_i^2} \ . \tag{19}$$

where $\text{vol}G$ is the volume of the graph (sum of degrees). In other words, for $K_N$, the learnable weights $\mathbf{w}_i$ mirror the Fiedler vector (they can be interpreted in this way). $\square$

**Corollary 2.** *The Barbell graph $B_{2N}$ of $2N$ nodes, is formed by linking two cliques of $N$ nodes each by a unique link which is viewed as a relaxed cut in IST.*

*Proof.* Given the link $i' \sim j'$ the edge that links the two cliques. Consider $i \sim j$ and *internal edge* $E_{int}$ in any of the two cliques if $j \neq i'$ (left clique) or $j \neq j'$ (right clique). Then $i \sim i'$ and $i \sim j'$ are called *external edges* $E_{ext}$.

Now, leveraging again Eq. 16 in Theorem 1 we have that for internal edges $i \sim j$ the corresponding term in the Fiedler equation is expanded as follows:

$$(f_i - f_j)^2 = (\mathbf{w}_j - \mathbf{w}_i)^2 \ . \tag{20}$$

However, for external edges, $j'$ and $i'$ are reachable from their opposite cliques. Then, defining $\Delta\mathbf{w}_i := \mathbf{w}_{i'} - \mathbf{w}_i$ we have

$$(f_i - f_{i'})^2 = [\Delta\mathbf{w}_i - \mathbf{w}_{j'}]^2 \ . \tag{21}$$

and similarly

$$(f_i - f_{j'})^2 = [\mathbf{w}_{j'} - \Delta\mathbf{w}_i]^2 \ . \tag{22}$$

Then, we expand $\sum_{i \sim j}(f_i - f_j)^2$ as follows:

$$\sum_{i \sim j \in E_{int}} (\mathbf{w}_i - \mathbf{w}_j)^2 + (\mathbf{w}_{i'} - \mathbf{w}_{j'})^2 + \sum_{i \sim j \in E_{ext}} (\Delta\mathbf{w}_i - \mathbf{w}_{j'})^2 \ . \tag{23}$$

Each clique has $N(N - 1)/2$ edges, half internal and half external. therefore, we have $N(N - 1)/2$ internal edges and $N(N - 1)/2$ external. Internal edges and the linking one behave as in the standard theory. However, the term corresponding to external edges includes the reaching of $i'$ and $j'$ from opposite cliques. This enforces the minimization of $(\Delta\mathbf{w}_i - \mathbf{w}_{j'})^2 = (\mathbf{w}_{i'} - \mathbf{w}_i - \mathbf{w}_{j'})^2$ which makes $i$ close to $i'$ (for $i$ in the left clique) and close to $j'$ (for $i$ in the right one).

As per the numerator of the Fiedler equation, we have that the degree of all nodes except $i'$ and $j'$ is $d_i = N - 1$, whereas $d_{i'} = d_{j'} = N$. Then, the denominator becomes

$$\sum_{i \neq i', i \neq j'} 2(N - 1)^2 \mathbf{w}_i^2 + N^2 \mathbf{w}_{i'}^2 + N^2 \mathbf{w}_{j'}^2 \ , \tag{24}$$

where all the magnitudes are $O(N^2)$ and the numerator dominates the minimization. $\square$

**Corollary 3.** *For the star graph $S_N$ with a central node $i_0$ linked to $N$ outer nodes $j$ not linked between them, then $\mathbf{w}_{i_0}^2 > \mathbf{w}_j^2 \; \forall j \neq i_0$.*

*Proof.* Instantiating Eq. 15 for this graph and considering that the peripheral nodes have unit degree, we obtain

$$\lambda_2 = \min_{f \perp \mathbf{D1}} \frac{\sum_{i_0 \sim j} (\mathbf{w}_j - N\mathbf{w}_{i_0})^2}{\sum_{j \neq i_0} \mathbf{w}_j^2 + N\mathbf{w}_{i_0}^2} \tag{25}$$

As we must maximize the denominator, the weight of the central node is larger than that of the peripheral ones. □

**Corollary 4.** *Path graph $P_N$ of $N$ nodes. Given the sorted nodes $i_1, i_2, \ldots, i_N$, for $1 < k < N$ we define the increments $\Delta\mathbf{w}_k := (\mathbf{w}_{i_k} - \mathbf{w}_{i_{k+1}})$. Then $\mathbf{w}_{i_1} < \mathbf{w}_{i_k}, k > 1$.*

*Proof.* Instantiating Eq. 15 and isolating terms, we discover that $\mathbf{w}_{i_1}^2$ must be minimal:

$$\lambda_2 = \min_{f \perp \mathbf{D1}} \frac{\mathbf{w}_{i_1}^2 + \sum_{i_k \sim i_{k+1}} (\Delta\mathbf{w}_k + \Delta\mathbf{w}_{k+1})^2 + \mathbf{w}_{i_{N-2}}^2}{\mathbf{w}_{i_1}^2 + \sum_{1 < i_k < N} 2\mathbf{w}_{i_k}^2 + \mathbf{w}_{i_N}^2} \; . \tag{26}$$

□

**Corollary 5.** *Cycle graph $C_N$. For $1 \leq k \leq N$ we define the increments $\Delta\mathbf{w}_k := (\mathbf{w}_{i_k} - \mathbf{w}_{(i_{k+1} \mod N)})$. Then all $\mathbf{w}_i^2$ have a similar magnitude.*

*Proof.* Now the last node $\mathbf{w}_{i_N}$ is linked with the first $\mathbf{w}_{i_1}$ and all the nodes have degree 2. Then

$$\lambda_2 = \min_{f \perp \mathbf{D1}} \frac{\sum_{i_k \sim i_{k+1}} (\Delta\mathbf{w}_k + \Delta\mathbf{w}_{k+1})^2}{\sum_{i \in V} 2\mathbf{w}_i^2} \; . \tag{27}$$

□

**Lemma 1.** *In IST, the condition $f \perp \mathbf{D1}$ is rewritten as $\mathbf{w} \perp \mathbf{d}_N$, where $\mathbf{d}_N(i) := \sum_{p \in N(i)} d_p$ is the* local volume *of $i \in V$ excluding $d_i$.*

*Proof.* The condition $f \perp \mathbf{D1}$ (Fiedler vector must be orthogonal to the degree vector) means $\sum_{i \in V} f_i d_i = 0$. Then, by rewriting the denominator in the spectral gap (see Theorem 1) we have

$$\begin{aligned}
\sum_{i \in V} f_i d_i &= \sum_{i \in V} [\sum_{p \in N(i)} \mathbf{w}_i] d_i \\
&= \sum_{i \in V} \mathbf{w}_i \sum_{p \in N(i)} d_p \\
&= \sum_{i \in V} \mathbf{w}_i \mathbf{d}_N(i) = 0 \; . 
\end{aligned} \tag{28}$$

Therefore, $\mathbf{w} \perp \mathbf{d}_N$ and $\lambda_2$ is rewritten as follows:

$$\lambda_2 = \min_{\mathbf{w} \perp \mathbf{d}_N} \frac{\sum_{i \sim j} [\sum_{p \in N(i)} \mathbf{w}_p - \sum_{q \in N(j)} \mathbf{w}_q]^2}{\sum_{i \in V} \mathbf{w}_i^2 \mathbf{d}_N(i)} \; . \tag{29}$$

As a result, the decision boundary provided by $\mathbf{w}$ must be orthogonal to the local volume not to individual degrees. □

**Lemma 2.** *The IST Harnack equality shows a principle for label propagation relying on weight neighborhoods.*

*Proof.* The Harnack equality shows that $f$ is Harmonic, i.e. given $\lambda_2$, we have that $f_i$ satisfies

$$\frac{1}{d_i} \sum_{j \in N(i)} (f_i - f_j) = \lambda_2 f_i \ . \tag{30}$$

Then, each component $f_i$ of the Fiedler vector is defined (up to the scale given by $\lambda_2$) as the average discrepancies between its neighbors.

Working on the above equation we obtain

$$\frac{f_i}{d_i} - \frac{\sum_{j \in N(i)} f_j}{d_i} = \lambda_2 f_i$$

$$\frac{f_i}{d_i} - \lambda_2 f_i = \frac{\sum_{j \in N(i)} f_j}{d_i}$$

$$f_i(1 - \lambda_2)d_i = \sum_{j \in N(i)} f_i$$

$$f_i = \frac{\sum_{j \in N(i)} f_j}{(1 - \lambda_2)d_i} \tag{31}$$

A straightforward translation to learnable weights yields

$$\sum_{p \in N(i)} \mathbf{w}_p = \frac{\sum_{p \in N(i)} \sum_{q \in N(p)} \mathbf{w}_q}{(1 - \lambda_2)d_p} \tag{32}$$

Now, suppose that $f_i = l_i, i \in V$ where $l_i \in \{-1, 1\}$ is a label. Then, if all the neighbors $q \in N(p)$ but $i$ are labeled (we denote it by $l_i = 0$) we have

$$\sum_{p \in N(i)} \mathbf{w}_p = \frac{\sum_{q \in N(p), p \in N(i)} l_q}{(1 - \lambda_2)d_i} \ . \tag{33}$$

Therefore, each label has a fractional contribution to the weights. This is the neural version of Laplacian learning. $\square$

**Theorem 2.** *Data labels lead to optimal partitions, but their transductive power decays with the shortest-path (SP) distance between labeled and unlabeled nodes.*

*Proof.* Suppose that $f_i = l_i, i \in V$ where $l_i \in \{-1, 1\}$ is a label. Let $P = \{x_0 = i, x_1, \ldots, x_N = j\}$ be the shortest path of length $L$ between $i$ and $j$. From Lemma 2, Eq. 31 results in

$$f_{x_1} = \frac{\sum_{p \in N(x_1)} l_p}{(1 - \lambda_2)d_{x_1}} \tag{34}$$

If all the nodes $k \neq i$ are unlabeled ($l_k = 0$, which indicates maximal uncertainty in the Fiedler partitioning), then we have the succession

$$f_{x_1} = \frac{1}{(1 - \lambda_2)d_{x_1}} \cdot l_i$$

$$f_{x_2} = \frac{l_i^2}{(1 - \lambda_2)^2 d_{x_2} d_{x_1}}$$

$$\vdots$$

$$f_j = \frac{l_i^L}{(1 - \lambda_2)^L \prod_{i=1}^{L} d_{x_i}} \tag{35}$$

which results in $f_j \to 0$ for a moderate $L$ even for a small degree (for instance degree 2 in a Path graph).

Translating the succession in Eq. 35 to the weights notation, we skip $x_1$ and get the label of $x_0$ when visiting $x_2$ (second-order neighbor). Then we have

$$\sum_{q \in N(j)} \mathbf{w}_q = \frac{l_i^{N-1}}{(1-\lambda_2)^{N-1} \prod_{i=2}^{N} d_{x_i}} , \tag{36}$$

with similar results. $\qquad \square$

**Corollary 6.** *The injection of a label $l$ at level $r$ may relax the decay if it is compatible with $l_i$ (same sign) or enforce it if it is not compatible.*

*Proof.* The status of a label at level $r$ can be modified by a single "informed" adjacent node:

$$f_{x_r} = \frac{l_i^r + l}{(1-\lambda_2)^r \prod_{i=1}^{r} d_{x_i}} . \tag{37}$$

$\qquad \square$

**Corollary 7.** *In general, we need both positive and negative labels to induce consistent partitions in the IST Fiedler vector.*

*Proof.* At this point, it is interesting to leverage Lemma 1 which states that the weights are orthogonal to local volumes, i.e.

$$\sum_{i \in V} \mathbf{w}_i \mathbf{d}_N(i) = 0 \tag{38}$$

Since, we have also $\mathbf{w} \neq \mathbf{0}$, at least one weight is negative.

However, if all our labels are positive, these negative weights come from flipping the sign of small labels of distant nodes or of close nodes with a very high degree, which results in uncertainty $f_i \approx 0$.

An exception to this rule is the star graph $S_N$ where the largest magnitude $\mathbf{w}_i$ is assigned to the central node. In this case, a single positive label is enough. $\qquad \square$

## C  PRACTICAL FINDINGS

The above results emerge from a blend of classical SGT and experimentation. Herein, we summarize our experiments when trying to set the *minimum number of labels needed to provide full accuracy* in several prototypical graphs.

**Barbell graph** $B_{2N}$ links two cliques of size $N$ with a single edge. Minimal labeling puts positive and negative edges at the extremes of the cutting edge. Local high density (large degree) in the clique's block label propagation but small SPs (unit length) make the difference.

When modifying the Barbell graph so that one community is "absorbed" by the other, we need only two more labels. Again, the unit length of SPs makes it work.

**Path Graph** $P_N$ suffers from label uncertainty for large values of N. We start by labeling the extremes of the central edge in the path as $(-1, +1)$. This is a good heuristic to set a "polarized edge": evaluate how powerful it is in terms of minimizing the NCut of the induced partition.

The central polarized edge at position $O(N/2)$ bisects the graph in two halves and depending on $N$ further labels are needed to bisect each half at positions $O(N/4)$ and $O(3N/4)$. In addition, two more labels are needed at the two extremes of the path.

**Cycle Graph** $C_N$ behavior is similar to $P_N$ with the "polarized edge" at $O(N/2)$. nodes $N$ and 1. Adding labels at $O(N/4)$ and $O(3N/4)$ we reach an accuracy of 92.5%.

**Star Graph**. In $S_N$, where half of the peripheral nodes and the central one belong to the same class, a single label placed at the central node yields full accuracy.

**Balanced Tree**. A balanced tree $B_{R,T}$ with branching factor $R > 1$ and $T$ levels has $N = R^T - 1$ nodes where $R^{T-1}$ are leaves (with unit degree) and the remaining interior nodes have degree $R + 1$). For $R = 2$ (binary) we have adopted the following labeling strategy: the root of each of the subtrees is labeled with opposite signs., and the root of the full tree (belonging to one of the classes) is not labeled. For $T = 2$ levels, we achieve an accuracy of $89\%$: a single subtree including leaves is misclassified. If in addition, we label correctly the first level of the subtrees we have full accuracy. This graph is interesting because it exemplifies the over-squashing issue.

**SBMs**. Stochastic Block Models, with probability $p = 0.75$ of intra-cluster linkage and probability $q = 0.25$ of inter-cluster linkage. This is a hard case where we want to test the IST cut relaxation. Having $O(N/3)$ samples (half in each cluster) we only achieve an accuracy of $50\%$. Setting now $p = 0.80$ and $q = 0.20$ we peak an accuracy of $90\%$ with $O(2N/3)$ labels.

# D    DATASET ANALYSIS AND EXPERIMENTAL SETUP

## D.1    DATASET STATISTICS

We present a comprehensive overview of the datasets utilized in our experiments, encompassing both node classification and graph classification tasks. Tables 5 and 6 provide detailed statistics for these datasets.

Table 5: Statistics of node classification datasets.

|  | Cornell | Texas | Wisconsin | Cora | Citeseer | Chameleon |
|---|---|---|---|---|---|---|
| #NODES | 140 | 135 | 184 | 2485 | 2120 | 832 |
| #EDGES | 219 | 251 | 362 | 5069 | 3679 | 12355 |
| #FEATURES | 1703 | 1703 | 1703 | 1433 | 3703 | 2323 |
| #CLASSES | 5 | 5 | 5 | 7 | 6 | 5 |
| DIRECTED | TRUE | TRUE | TRUE | FALSE | FALSE | TRUE |
| HOMOPHILY | 0.11 | 0.30 | 0.21 | 0.81 | 0.74 | 0.23 |
| AVG DEGREE | 1.77 | 1.62 | 2.05 | 3.89 | 2.73 | 15.85 |
| DENSITY | 0.009 | 0.008 | 0.008 | 0.014 | 0.008 | 0.007 |

Table 6: Characteristics and Statistics of eight graph classification datasets.

| Classification | Datasets | #Graphs | Avg Nodes | Avg Edges | Classes |
|---|---|---|---|---|---|
| Biological | PROTEINS | 1,113 | 39.06 | 72.82 | 2 |
|  | ENZYMES | 600 | 32.63 | 62.14 | 6 |
| Chemical | MUTAGENICITY | 4,337 | 30.32 | 30.77 | 2 |
|  | MUTAG | 188 | 17.93 | 19.79 | 2 |
|  | BZR | 405 | 35.75 | 38.36 | 2 |
|  | PTCMM | 336 | 13.97 | 14.32 | 2 |
| Social | COLLAB | 5000 | 74.49 | 2457.78 | 3 |
| Networks | IMDB-BINARY | 1000 | 19.77 | 96.53 | 2 |

For node classification datasets (Table 5), we report additional metrics such as homophily, average degree, and density. These metrics provide insights into the structural properties of the networks. Homophily indicates the tendency of nodes to connect with others of the same class, average degree shows the typical number of connections per node, and density reflects the overall connectedness of the graph.

Graph classification datasets (Table 6) are categorized into biological, chemical, and social network domains. We present the total number of graphs, the average number of nodes and edges per graph, and the number of classes for each dataset.

## D.2 EXPERIMENTAL ENVIRONMENT

All experiments were conducted using the hardware specifications outlined in Table 7. Concerning software, we have used PyTorch Geometric (PyG), NetworkX and scikit learn as main Python libraries.

Table 7: Hardware specifications for experimental setup.

| Component | Specification |
|---|---|
| CPU | AMD 7742 64-Core @ 2.25 GHz |
| GPU | NVIDIA A100 Tensor Core (40GB VRAM) |
| RAM | 1024GB DDR4 |
| Storage | 2TB NVMe SSD |
| Operating System | Ubuntu 20.04.5 LTS |

