# OpenReview forum: "Learnable Eigenfunctions for Graph Rewiring: Balancing Local Community Structure and Global Connectivity"
_ICLR.cc/2026/Conference — ICLR 2026 Conference Withdrawn Submission_

### Official Review · Reviewer_vuq9 · 2025-10-24

**Soundness:** 2
**Presentation:** 3
**Contribution:** 1
**Rating:** 2
**Confidence:** 4

**Summary:**

This paper provides a learnable graph rewiring strategy that combines minimizing a Dirichlet energy of the graph(s) with the optimization loss of the task, creating a "task-reactive consensus eigenspace" embedding. They choose to add edges at two scales, intra and inter community-wise. With their method, they trade higher method complexities for an optimization process. They also (orthogonally) suggest to use rewiring methods as augmentations for graph classification.

**Strengths:**

- The idea of rewiring-based data augmentation for graph classification is interesting, as it improves on the random-rewiring augmentation. Although it would need more exploration with other methods and more theory on the quality of diversity.
- There is good clarity of the method in terms of step by step explanation, and the paper is well-written.

**Weaknesses:**

1. Data augmentation is orthogonal to rewiring in general, and your method in particular. You should either focus on exploring the benefits of rewiring-based augmentation in general (for all rewiring methods), or use it as an ablation (for all rewiring methods) that is not central to your arguments. In both cases, using augmentation for more methods should be due.
2. There are already rewiring methods that, even without training, explore the interplay between communities and task (1), so (i) the novelty on local versus global (intra/inter communities) and task versus structure rewiring is lost; (ii) their node classification results seem to be better than yours, so a comparison is due.
3. Because this rewiring is learned and not pre-processed, comparisons to other graph structure learning (GSL) methods or rewiring-learning methods (2) might be due as well.
4. There is already a fast eigenvalue approximation based rewiring in linear time in (3), so (i) novelty of method complexity is lost; and (ii) doing it in a learnable way is not justified, at least not by over-squashing alone.
5. The claim that the method balances "local and global" is not truly fulfilled, as you arbitrarily choose to add half and half of each kind. Therefore, you don't really have a way of integrating them together, or not more than other works in the literature.

(1) GNNs Getting ComFy: Community and Feature Similarity Guided Rewiring. Celia Rubio-Madrigal et al. ICLR 2025.

(2) Probabilistically Rewired Message-Passing Neural Networks. Chendi Qian et al. ICLR 2024.

(3) Spectral Graph Pruning Against Over-Squashing and Over-Smoothing. Adarsh Jamadandi et al. NeurIPS 2024.

**Questions:**

1. Over-squashing is not defined to be "reactive to labels", or, technically, by long-range interactions (4). Then what is it that the method is solving, that other methods are not?
2. Can the data augmentation be done for all the rewiring methods or does it only work for yours and why?
3. What does it mean that the method increases robustness? There needs to be support for this, either by ablations of noise or out-of-distribution or elsewhere.
4. Why does the "consensus eigenspace" for all graphs in the graph classification setting exist and is an informative object? Why is it needed for the tasks? Why is it better to share among graphs than not to?
5. What is the concept of "data-centered eigenvalues" of the adjacency matrix? As far as I'm concerned, this doesn't exist.
6. Does adding the task-dependent loss to the rewiring loss make the optimization harder?
7. How do you "maintain graph sparsity" if you are adding edges?
8. How does the choice of K affect performance? How much of the gains come from taking into account more eigenvalues, and would this be introducible to other methods?
9. How does deleting edges improve performance instead of adding? Given that (3) finds that removing edges already improves over-squashing and over-smoothing together.
10. What does it mean that your method "introducing edges that enhance structural cohesion and optimize information flow"? Don't all rewiring methods do this already? In fact, I'm sure that FoSR and GTR don't "add edges indiscriminately"...

(4) Oversmoothing, Oversquashing, Heterophily, Long-Range, and more: Demystifying Common Beliefs in Graph Machine Learning. Adrian Arnaiz-Rodriguez & Federico Errica.

---

### Official Review · Reviewer_x7QE · 2025-10-27

**Soundness:** 3
**Presentation:** 3
**Contribution:** 2
**Rating:** 6
**Confidence:** 4

**Summary:**

The paper proposes Inductive Spectral Theory (IST), a learnable, label-aware graph rewiring and data augmentation framework for GNNs. The core idea is to replace fixed spectral objects (eigenvectors/Fiedler vector, spectral gap) with learned eigenfunctions produced by an MLP that minimizes Dirichlet energy plus a supervised task loss. These learned embeddings are then used to (1) estimate pairwise node distances; (2) add new “local” edges between close nodes (to strengthen community structure and reduce over-smoothing); and (3) add “global” edges between far nodes (to relieve over-squashing and improve long-range information flow); and (4) generate augmented training graphs by this rewiring strategy.

**Strengths:**

1. Instead of computing classical Laplacian eigenvectors (Fiedler vector, etc.), IST learns eigenfunction-like embeddings via an MLP trained to minimize a Dirichlet-style smoothness objective plus the downstream task loss. This tries to transfer supervision into the “spectral space,” making the rewiring label-aware.
2. The paper distinguishes “local edges” (short learned-distance edges, encouraging tighter communities) from “global edges” (long learned-distance edges, encouraging long-range communication). This mirrors the known trade-off in the literature: locality-preserving rewiring like LASER aims to maintain sparsity and local structure while still reducing over-squashing。
3. On node classification benchmarks (Cora, Citeseer, Chameleon, etc.) and graph classification benchmarks (MUTAG, IMDB-B, ENZYMES, etc.), IST usually reports the best accuracy among listed baselines

**Weaknesses:**

1. DiffWire already frames rewiring as a learned, differentiable, inductive procedure that optimizes spectral / topological objectives, rather than relying on a fixed graph. This means the idea of “learn the rewiring policy jointly with the task, in an inductive way” is not that novel. [1]

2. Theoretically, the paper gives no quantitative analysis of the newly added edges: to what extent do they improve connectivity or preserve sparsity? Empirically, prior rewiring work typically *measures* structural effects — e.g., reduced over-squashing via effective resistance / curvature, increases in spectral gap, reduced graph diameter, or preserved locality — and uses those metrics to argue it actually fixed known bottlenecks like over-squashing and over-smoothing. [1][2][3][4]
   In contrast, IST mostly reports classification accuracy. It does not report curvature changes, effective resistance, spectral gap increases, diameter shrinkage, clustering quality, or sparsity metrics before vs. after rewiring, so the causal story (“we balance local cohesion and global connectivity”) is asserted but not quantitatively demonstrated.

3. The paper claims scalability and inductive generalization, but presents no experiments on truly large graphs. There are also no runtime / memory benchmarks to support the claimed complexity advantages over eigendecomposition-heavy spectral methods or curvature-based rewiring.

4. The optimization in Eq. (7) includes an orthogonality constraint on the learned eigenfunction basis, but the paper does not explain how that constraint is actually enforced under SGD.

5. Key algorithmic pieces (orthogonality enforcement, the distance-based local/global edge sampling policy, and the padding / batching scheme for multi-graph training in graph classification) are not described precisely enough for a reader to reimplement without guesswork.

---

### References

[1] DiffWire. Differentiable, inductive graph rewiring that learns commute-time / spectral-gap style signals and updates edges jointly with the downstream objective, emphasizing scalability and generalization to unseen graphs.

[2] FoSR. Rewiring via spectral expansion to alleviate over-squashing, with an architecture designed to avoid over-smoothing; evaluates improvements using spectral / connectivity diagnostics, not just accuracy.

[3] BORF. Curvature-driven rewiring using Ollivier–Ricci curvature, linking negative curvature to over-squashing and positive curvature to over-smoothing, and reporting curvature / effective-resistance style metrics.

[4] LASER. Locality-aware sparsity-preserving rewiring that explicitly targets (i) reducing over-squashing, (ii) preserving locality/communities, and (iii) keeping graphs sparse, and validates those properties quantitatively.

**Questions:**

See Weaknesses

---

### Official Review · Reviewer_5Ki8 · 2025-10-31

**Soundness:** 2
**Presentation:** 2
**Contribution:** 2
**Rating:** 2
**Confidence:** 4

**Summary:**

The paper proposes a new framework, Inductive Spectral Theory (IST), for graph rewiring to improve Graph Neural Network (GNN) performance. The central idea is to learn spectral embeddings using a data-driven approach. This is implemented via a Multi-Layer Perceptron (MLP) that learns node representations by optimizing a combination of the Dirichlet energy and a task-specific loss, making the learned "eigenfunctions" reactive to labels. The resulting node embeddings are used to compute a distance distribution. The graph is then rewired by adding a fixed number of "local" edges (for pairs with small distances) and "global" edges (for pairs with large distances). The authors claim this method balances local community structure and global connectivity, thereby mitigating both over-smoothing and over-squashing. This rewiring is also presented as a principled data augmentation technique. Empirical results on several node and graph classification benchmarks show performance improvements over baseline GNNs and other rewiring/augmentation methods.

**Strengths:**

S1. The core motivation—to create an "inductive" and "learnable" version of spectral graph theory—is compelling. Moving beyond static, pre-computed spectral properties to a framework that adapts eigenfunctions based on task-specific labels is an interesting research direction.

S2. The paper correctly identifies major challenges in GNNs (over-squashing, over-smoothing) and spectral methods (scalability) as its targets.

S3. The empirical evaluation is reasonably broad, covering both node and graph classification tasks and comparing against a relevant set of existing rewiring and augmentation methods.

**Weaknesses:**

W1.The paper's central claims and contributions are confused.
The core contribution is unclear. The method is described as learning eigenfunctions, but the implementation appears to be an MLP that learns a node embedding, $f(A)$, by minimizing a loss (Eq. 7). It is not clear how this is fundamentally superior to many standard GNN embedding techniques. The subsequent rewiring—adding edges based on a distance threshold in this embedding space—seems popular in graph rewiring methods.
The paper claims to solve over-squashing, over-smoothing, and data augmentation. However, it never specifies in detail how IST is responsible for which benefit. Is over-smoothing solved by adding local edges? Is over-squashing solved by adding global edges? How does the loss function balance this? The connection is assumed, not explained.
The paper frequently and confusingly switches between node classification and graph classification. Figure 2, the primary methodological illustration, depicts a node classification task (labeled vs. unlabeled nodes). However, Section 4.2 immediately describes the methodology for graph classification. These are distinct problem settings, and the paper fails to explain how the method is adapted between them.

W2. Several key technical claims are made without sufficient explanation or evidence.
The paper repeatedly claims to "facilitate long-range connections" and "preserve both label information and structural properties". These mechanisms are not explained. How does the model ensure long-range connections are added in a principled way (beyond just sampling from pairs with > median distance)? How does the loss in Eq. 7 truly "preserve" structural properties (via Dirichlet energy) while also being "reactive" to labels?
The caption for Figure 2 claims that " over-squashing "is avoided by clustering the node embeddings. This is an extraordinary claim that is left entirely unexplained. Over-squashing is a problem of information loss from distant nodes (a global problem), while clustering emphasizes local structure. This claim seems contradictory and is not defended in the text.
The rewiring step (Sec 4.2, step 4) is poorly defined. How is the "distribution of distances" $\mathcal{D}_i$ created? Is this from all $O(N^2)$ pairs, which is computationally infeasible for large graphs? Or from samples? Why is the median chosen as the threshold? This seems arbitrary. Furthermore, no analysis of this learned distance distribution is ever shown in the experiments.

W3. The paper's components are not well-aligned.
Figure 1 is described as an "analysis" but is a vague illustration that provides no insight into how the four methods work or mitigate bottlenecks. It fails to support the claims in Table 1 (e.g., that FoSR/GTR do not preserve locality).
The paper's core objective function, Eq. 7, is presented with unexplained $\mathcal{L}$.

W4. The theoretical analysis in the appendix is based on a linear mapping ($f_i = \langle w, a_{:i} \rangle$). However, the actual method uses a non-linear $MLP_{\theta}(A)$. The paper does not justify how the insights from the linear analysis (e.g., Theorem 1) apply to the non-linear model.

**Questions:**

Q1. Please clarify the objective function in Equation 7.

Q2. In Table 1, what does "Differentiable" refer to?

Q3. How is the methodology adapted between the node classification setup (used in Figure 2) and the graph classification setup (described in Sec 4.2)?

Q4. Please explain the claim in the Figure 2 caption: "over-squashing ... is avoided by clustering the node embeddings."

Q5. What is the computational process for creating the "distribution of pairwise distances" $\mathcal{D}_i$? Is this an $O(N^2)$ computation? If not, how are node pairs sampled?

Q6. How does the theoretical analysis of a linear model in the appendix provide valid insights for the non-linear MLP-based method used in the experiments?

Q7. Given that the ablation study (Table 4) shows the optimal local/global balance is dataset-dependent, why does the proposed method use a fixed "N/2 local, N/2 global" strategy? How can the method be improved to learn this balance?

---

### Official Review · Reviewer_G96A · 2025-11-01

**Soundness:** 2
**Presentation:** 2
**Contribution:** 2
**Rating:** 2
**Confidence:** 4

**Summary:**

This paper proposes Inductive Spectral Theory (IST), aiming to learn empirical eigenfunctions reactive to data and labels for graph rewiring and data augmentation in GNNs. The goal is to mitigate over-squashing and over-smoothing by adding local and global edges based on learned spectral representations.

**Strengths:**

(1) Interesting and ambitious idea to make spectral graph quantities learnable.

(2) Motivation linking spectral theory and graph rewiring.

(3) The notion of label-reactive eigenfunctions is novel.

**Weaknesses:**

(1) Theoretical inconsistency: The core equation (Eq. 7) defines function f based on a fixed Laplacian, so no actual rewiring occurs—contradicting the claimed contribution.

(2) Terminological confusion: The so-called “empirical eigenfunctions” are just learned node embeddings, not eigenvectors of Laplacian matrix after rewring.

(3) Label leakage: Labels are fed into the eigenfunction learning stage, invalidating inductive evaluation.

(4) Permutation sensitivity: Flattening adjacencies into an MLP input breaks permutation invariance.

(5) Algorithm–theory mismatch: The pipeline steps in section 4.2 conflict with the objective in Eq.7; Laplacian is fixed while edges are added later without consistency.

(6) Weak justification: Edge addition rules (local/global) are heuristic and lack spectral grounding.

**Questions:**

(1) In section 4.2, please justify the use of a standard MLP on the padded, permutation-variant adjacency matrix to derive the EEs. How is permutation invariance guaranteed?

(2) Could the authors clarify the theoretical justification for minimizing the Dirichlet energy with respect to the original Laplacian $\mathcal{L}$, when the final GNN is trained on the significantly different rewired graph in step 4 in section 4.2?

**Details Of Ethics Concerns:**

no.

---

### Note · Authors · 2025-12-03

I have read and agree with the venue's withdrawal policy on behalf of myself and my co-authors.